# The Formation, Structural Characteristics, Absorption Pathways and Bioavailability of Calcium–Peptide Chelates

**DOI:** 10.3390/foods11182762

**Published:** 2022-09-08

**Authors:** Jiulong An, Yinxiao Zhang, Zhiwei Ying, He Li, Wanlu Liu, Junru Wang, Xinqi Liu

**Affiliations:** 1National Soybean Processing Industry Technology Innovation Center, Beijing Technology and Business University (BTBU), Beijing 100048, China; 2Beijing Advanced Innovation Center for Food Nutrition and Human Health, Beijing Technology and Business University (BTBU), Beijing 100048, China

**Keywords:** calcium-peptide chelate, calcium supplement, characterization, transport pathway, bioavailability

## Abstract

Calcium is one of the most important mineral elements in the human body and is closely related to the maintenance of human health. To prevent calcium deficiency, various calcium supplements have been developed, but their application tends to be limited by low calcium content and highly irritating effects on the stomach, among other side effects. Recently, calcium–peptide chelates, which have excellent stability and are easily absorbed, have received attention as an alternative emerging calcium supplement. Calcium-binding peptides (CaBP) are usually obtained via the hydrolysis of animal or plant proteins, and calcium-binding capacity (CaBC) can be further improved through chromatographic purification techniques. In calcium ions, the phosphate group, carboxylic group and nitrogen atom in the peptide are the main binding sites, and the four modes of combination are the unidentate mode, bidentate mode, bridging mode and α mode. The stability and safety of calcium–peptide chelates are discussed in this paper, the intestinal absorption pathways of calcium elements and peptides are described, and the bioavailability of calcium–peptide chelates, both in vitro and in vivo, is also introduced. This review of the research status of calcium–peptide chelates aims to provide a reasonable theoretical basis for their application as calcium supplementation products.

## 1. Introduction

Calcium is the most abundant mineral element in the human body, representing ap-proximately 1.5–2% of the weight of a normal human body. Most calcium is in the form of phosphate in bones and teeth, while the remainder takes the form of ions in soft tissues, cells and blood [1,2]. Calcium performs a variety of functions in the body, including the formation of bones and teeth, the linking of nerve excitation and muscle contraction and the regulation of enzyme activity. Long-term calcium deficiency in the body can lead to cramps, osteoporosis and rickets [3,4,5]; therefore, various calcium supplements such as calcium salts, organic calcium salts and amino acid chelate calcium have been developed to ensure normal calcium levels [6]. Although calcium supplementation does provide certain benefits, it is associated with unwanted problems such as intense irritation, related adverse reactions and the hampering of amino acid absorption that have yet to be effectively solved [7,8,9]. Therefore, a new calcium supplement is required that can provide superior biocompatibility and obvious calcium supplementation with no adverse side effects.

Numerous studies have shown that peptides can provide a variety of biological benefits, including antioxidant and anti-inflammatory activities, fatigue relief and the improvement of intestinal flora. Compared with other functional ingredients, peptides have a higher safety profile, they are hypoallergenic and their production is relatively cost-effective. FDA regulations give protein hydrolysates containing bioactive peptides the status of Generally Recognized as Safe (GRAS) food substances [10]. The absorption of peptides has the characteristics of low energy consumption, less saturable carriers and rapidity [11,12]. Moreover, some peptides are able to bind with calcium ions, thus forming calcium–peptide chelates [13,14], which in-depth cellular and animal experiments have demonstrated to show potential for application as calcium supplements. Herein, we describe the preparation and structural characteristics of calcium–peptide chelates and evaluate their bioavailability.

## 2. Sources of Calcium-Binding Peptides

Calcium-binding peptides (CaBP) can be isolated from a variety of sources, which can be summarized into two categories, namely plant sources and animal sources, some of which are discussed below and listed in Table 1. The calcium load of the peptide fragment ranges from 10–950 μg/mg according to the available studies, with the highest load up to 943.60 μg/mg.

### 2.1. Plant Proteins

Plant proteins have the advantage of wide availability as comprehensive amino acids that are easily absorbed by the human body. Studies have shown that CaBP can be extracted from plant proteins. Soy protein, for example, is rich in aspartic acid and glutamic acid, both of which have been proven to have certain calcium-binding capacities (CaBC) [30]. Lv et al. isolated CaBP via the hydrolysis of soybean protein and demonstrated CaBC of up to 57.25 μg/mg [31]. It has also been reported that three CaBPs, with the sequences LLLGI, AIVIL and HADAD, were extracted from mung bean protein hydrolysate. LLLGI was found to have the highest CaBC at 943.60 μg/mg, which suggested that the presence of leucine or isoleucine at the C-terminal or N-terminal may be helpful in enhancing the CaBC of a peptide [21]. Based on Liu’s finding, CaBP could also be made from wheat germ proteins with a CaBC of 67.5 μg/mg [32]. Plant seeds are also widely reported to contain a variety of active substances. Wang et al. extracted a peptide with a CaBC of 191.5 µg/mg from cucumber seeds, which the authors asserted could be used as a new calcium supplement [17]. Furthermore, two CaBP with the sequences of AFNRAKSKALNEN and YDSSGGPTPWLSPY were isolated from lemon basil seeds, with CaBCs of 278.14 µg/mg and 151.88 µg/mg, respectively [18]. *Chlorella* is a single-celled green algae that comprises more than 50% protein. After the extraction and hydrolyzation of *Chlorella pyrenoidosa* protein, Jeon et al. identified a peptide with a CaBC of 211 µg/mg, the sequence of which was NSGC, thereby demonstrating that serine, glycine and cysteine residues are the main amino acids responsible for calcium binding [15]. *Schizochytrium* sp. is another marine fungus with a large number of bioactive substances such as unsaturated fatty acids, pigments and proteins [33]. Cai et al. prepared a CaBP with the sequence FY from the *Schizochytrium* sp. protein and found that the CaBC of the peptide was 128.77 μg/mg [16]. Further studies have also found FY to be effective in improving calcium uptake by cells.

### 2.2. Animal Proteins

Casein, which constitutes 76–86% of the total protein in milk, is a common phosphorylated protein consisting of α_S1_-casein, α_S2_-casein, β-casein and κ-casein. The phosphorylation of proteins occurs mainly on serine, and phosphoserine can act as binding sites for calcium ions [34]. Casein phosphopeptides formed by the decomposition of pepsin and trypsin are the most common CaBPs [35,36]. In addition to milk, egg yolk also contains a large amount of phosphoprotein, and according to relevant research, phosphopeptides made from egg yolk fermentation have higher CaBCs than that of unfermented yolk [37]. Additionally, edible fish is rich in nutrition, with low fat and high protein content, and is a superior raw material for the production of bioactive peptides. For example, two tilapia peptides with the sequences of YGTGL and LVFL were prepared by Liu et al., who found that their CaBCs reached 76.03 μg/mg and 79.50 μg/mg, respectively [38]. Furthermore, Zhang et al. isolated a new decapeptide with a CaBC of 0.43 μg/mg from Pacific cod fish bone [39].

The high-value reuse of by-products from the processing of animal products is a popular topic of research. Some by-products can be prepared into bioactive peptides as they still have a large amount of protein. Animal bones are the major by-product of processed meat products and can be used as raw material for the preparation of bioactive peptides due to their collagen content. In relevant research, collagen peptides were extracted from sheep bone and were found to reach a CaBC of 56.39 μg/mg [26]. Furthermore, Zhang et al. determined the optimal preparation conditions of CaBP in bovine collagen with a single-factor experiment and obtained a maximum CaBC of 42.70 μg/mg [40]. A CaBP with the sequence VSGVEDVN was also isolated from pig plasma proteins, during which aspartic acid, glutamic acid and glycine residues played an important role in the calcium binding [27]. CaBPs have also been produced from other marine by-products such as sea cucumber eggs, snapper scales and octopus trimmings [41,42,43].

Thus, many peptides with different sequences have proven bindable to calcium ions; however, the common characteristics of their sequences have not yet been ascertained and warrant future research.

## 3. Preparation of Calcium Chelating Peptides

### 3.1. Hydrolysis

Bioactive peptides of different molecular weights are most commonly obtained via proteolysis or fermentation, and microorganisms produce proteases to break down the proteins in a substrate to form peptides. In the research of Wang et al., the CaBP was extracted from cucumber seeds via fermentation with *Bacillus subtilis* [17]. Sun et al. prepared a CaBP using bovine bone meal and soybean meal as the raw materials under the fermentation of *Bacillus natto*, with the CaBC reaching 707.47 ± 32.16 μg/mg under optimal conditions, determined via response surface methodology [44]. The combination of enzymatic digestion and fermentation has also been reported to achieve more thorough hydrolysis. For example, when a combination of enzymatic digestion and lactic acid bacteria fermentation was used to prepare CaBP from sheep bone collagen, collagen peptides containing a high level of free calcium were obtained, and a calcium–peptide chelate was prepared by precipitation using ethanol. This research thus also provided a novel and efficient use for sheep bones [26]. While fermentation methods do come at a lower cost, their yield of active substances tends to be lower, with poor specificity and a longer process [45].

In addition to the fermentation method, enzymatic digestion is also widely used for the preparation of CaBP due to its advantages of mild and controlled reaction conditions, high feasibility and safety [46]. The type of enzyme, length of hydrolysis and ratio of enzyme to substrate all affect the CaBC of the final product, and CaBP can be extracted from various sources of proteins via different types of proteases. For example, a CaBP with the sequence of YEIPAEDL was prepared via the hydrolysis of Japanese hairy shrimp protein extract using flavor enzymes [29]. Similarly, papain was used to hydrolyze tilapia collagen, and three CaBPs were obtained with the sequences GPAGPHGPVG, FDHIVY and YQEPVIAPKL [47]. In addition, pepsin, alkaline protease, fig protease and peptone have also been used in a number of studies to successfully hydrolyze proteins [20,22,48,49]. Research has also shown that the CaBC of hydrolysate is affected by enzymatic hydrolysis reaction conditions. Sun et al. hydrolyzed sea cucumber extracts with trypsin, papain, neutral protease, alkaline protease and flavoring protease, of which trypsin was found to be most effective, with the CaBC of tryptic hydrolysates significantly (90%) stronger than that of the lowest group [50]. According to research by Charoenphun et al., the CaBC of a tilapia protein peptide increased initially and then decreased in correlation with the de-grees of hydrolysis, with its CaBC highest when hydrolysis was 27.7% [23]. In a study about the effects of enzyme–substrate ratios on CaBC, the results showed that the CaBC of a corresponding hydrolysate increases as the ratio increases [35]. These findings show that by optimizing the type of enzyme, dosage and other factors, peptides with high CaBC can be obtained for use in subsequent research.

Several studies have shown that two or more proteases can improve hydrolysis capacity. Budseekoad et al. selected an alkaline protease, Flavourzyme, trypsin, pancreatic enzyme, pepsin, a mixture of pancreatic enzyme and alkaline protease (1:1) and a mixture of pancreatic enzyme and Flavourzyme (1:1) for hydrolysis [21]. Compared with the single enzyme, the mixed enzymes reached the same degree of hydrolysis more rapidly. In another study, whey protein was hydrolyzed via a combination of Flavourzyme and peptone, resulting in the extraction of a peptide (sequence EG) with a strong CaBC [51].

While fermentation and enzymatic methods are the traditional approaches in the production of various bioactive functional peptides, attention is increasingly being paid to new methods of production, such as high hydrostatic pressure processing and ultrasound-assisted extraction. However, these new techniques are currently mainly applied to the extraction and performance optimization of antioxidant and anti-inflammatory peptides [52,53]. Novel techniques that can contribute to the extraction of CaBP may therefore be an important direction for future research.

### 3.2. Purification

Further purification can improve the CaBC of peptides. Chromatography is a common method for the purification of active peptides, including immobilized metal affinity chromatography (IMAC), ion-exchange chromatography (IEC), hydroxyapatite affinity chromatography (HAC), gel-filtration chromatography (GFC) and reversed-phase high-performance liquid chromatography (RP-HPLC).

The principle of IMAC is that metal ions are bound to the stationary phase by means of ligands, and the metal can then bind to target molecules such as peptides and amino acids [54]. IMAC has many advantages, such as a stable stationary phase, high protein loading, mild elution conditions, easy regeneration of the column and low cost. IMAC has, for example, been used to purify CaBP from soybean hydrolysates and, in another study, from Alaskan cod skin [19,55].

IEC enables separation based on differences in electrical charge. This technique is widely used for the separation of peptides and proteins due to its biocompatibility and high resolution [56]. Relevant research has reported the extraction of CaBPs from wheat germ and from egg phosphoproteins using IEC [48,57].

HAC can specifically bind CaBP due to the presence of calcium ion groups on the stationary phase [46]. Peptides with high CaBC have been isolated from Alaska Pollack backbone; Pacific cod bone and tilapia scales using HAC techniques [22,58,59].

IMAC and HAC are usually used in the first step of purification, while GFC is usually used for the second step. GFC is a chromatographic method that achieves separation based on molecular weight. The operating conditions of this method are relatively mild, and the gel used for the chromatography is an inert material that can ensure the stability of the physical and chemical properties of the separated components. In related research, GFC was used to extract a CaBP from *Schizochytrium* sp. protein. The elution curve showed that the GFC divided the hydrolysate into three fractions, with the highest CaBC obtained from F3, the smallest of the molecular weight fractions [16].

RP-HPLC is based on the polarity of the substance and is usually used as the final step in the purification of CaBP due to its wide applicability, high separation efficiency and high sensitivity [46]. In a recent study, RP-HPLC was used to extract CaBP with the sequence EPAH from the Auxis thazard protein, and it was found that the CaBC of the peptide improved with each passage through the column during purification [25]. In another recent study, HAC, GFC and RP-HPLC were used to separate tilapia scale protein hydrolysates successively. The fractions with the highest CaBC, namely F3, F31 and F312, were selected after each purification operation, with their CaBCs detected as 104.00 μg/mg, 113.6 μg/mg and 116.4 μg/mg, respectively [59]. Similarly, the CaBC of phosphoprotein hydrolysate purified by IEC, GFC and RP-HPLC successively also significantly improved [57].

## 4. Calcium Binding Sites on Peptides

According to previous studies, the binding of peptides to calcium ions results in the formation of new internal structures, indicating the formation of new substances [40,48]. The binding of calcium to peptide relies primarily on the active site on the peptide chain, as has been determined via several techniques such as Fourier infrared spectroscopy (FTIR), X-ray photon spectroscopy (XPS) and nuclear magnetic resonance (NMR). The intermolecular forces between peptides and calcium ions are generated through hydrogen bonds, electrostatic interaction, coordination bonds and hydrophobic interaction. Sun et al. found that electrostatic interaction occurred between a krill peptide and calcium ions by measuring their thermodynamic parameters via ITC [60]. In calcium–peptide chelates, the oxygen and nitrogen atoms of the peptides are used mainly as electron donors, while the metal ions are used mainly as electron acceptors in the formation of coordination bonds [61,62]. After comparing the zeta potential of a peptide before and after calcium ion binding, Lin et al. observed that its absolute value was significantly reduced and that the potential of the chelate was lower than the peptide at the same pH, suggesting that the peptide and calcium ions were bound through the coordination bond [42]. Furthermore, Zhang et al. treated calcium–peptide chelates with dissociating agents with different sites of action and found that the chelates disintegrated when the energetic hydrophobic interactions and hydrogen bonds were disrupted [63]. This result confirmed that the formation and stabilization of chelates are dependent on the presence of hydrogen bonding and hydrophobic interactions. According to available studies, the main binding active sites on a peptide are its phosphate group, carboxy-oxygen end and nitrogen end [64].

### 4.1. Phosphate Group

Phosphorylation modification contributes to the binding of calcium ions to peptides, and phosphoserine is the main binding site for calcium ions [65]. Many casein phosphopeptides share an identical sequence, namely, Ser(P)-Ser(P)-Ser(P)-Glu-Glu, which has been identified as a binding site for metal ions [66].

In a recent study, Jiang et al. determined that the appropriate amount of phosphoryl groups in the peptide was a key factor in the CaBC of phosphopeptides and the dissolution of calcium from phosphate precipitates. Calcium retention has been found to be best in phosphopeptides with a phosphate retention of 35% [67]. Another study investigated the relationship between the molecular structures of phosphopeptides and their CaBCs by synthesizing 6 different phosphopeptides using 0–3 contiguous or discontinuous Ser(P)s. It was found that the greater the number of phosphoserines, the stronger the CaBC, and the CaBCs of peptides with nonadjacent phosphoserines were higher than that of those with adjacent phosphoserines [64]. Phosphorylation is a common method for protein modification, and current studies have shown significantly higher CaBCs in phosphorylated proteins. For example, the CaBC of peptides extracted from herring eggs increased from 68.8 mg/μg to 96.82 mg/μg after phosphorylation [68]. Similarly, Lee et al. phosphorylated soybean peptides and found a positive correlation between phosphorylation and CaBC [69].

### 4.2. Carboxylate Oxygen Atoms

Many peptides without a phosphate group have also been proved to have high CaBC, mainly accredited to the carboxyl terminus of the amino acids in the peptide [11]. Bao et al. measured the CaBC and carboxyl content of soybean protein hydrolysates and found a trend of positive correlation between the two indicators [70]. In that study, calcium ions were found to bind mainly to two oxygen atoms in the carboxyl group, namely, carbonyl oxygen and hydroxyl oxygen. Wang et al. analyzed the functional group changes in black bean peptides after binding to calcium ions by FTIR. The shift of the carboxylic acid group (-COO-) at 1450.27 cm^−1^ to a lower wave number suggested to the authors that -COO-Ca had been formed. Furthermore, a clear red shift was also seen in the range 1551.88 cm^−1^ to 1547.93 cm^−1^, thus confirming the involvement of the carbonyl group (C=O) in the binding process. These two conclusions demonstrated the combination of hydroxyl oxygen and carboxyl oxygen with the calcium ions [20]. In another assay, the infrared spectra of lemon basil seed peptides revealed that the amide I band (which referred to the carbonyl group absorption peak) shifted from 1624.34 cm^−1^ to 1626.85 cm^−1^ after binding to calcium ions, and a new peak appeared in the range of 1500–1400 cm^−1^ after chelation, suggesting that both oxygen atoms of the carboxyl group were involved in the binding [18]. Liu et al. found that the infrared spectra of wheat peptides bound to calcium ions was shifted in the region of 1658 cm^−1^ to 1650 cm^−1^ by C=O stretching vibrations, thus proving that the carbonyl group had combined with the calcium ions. The peak at 1402 cm^−1^ indicated that -COO- had shifted to 1414 cm^−1^ after binding, which showed the involvement of hydroxyl oxygen in the binding of the metal ion [32]. A binding site can also be determined using XPS. Following the examination of the O1s spectra of a peptide segment with sequence QEELISK and its chelate with calcium via XPS, Sun et al. observed that the three characteristic peaks in the spectrogram of the peptide segment, at 530.9 eV, 531.9 eV and 532.8 eV, corresponded to O-H, C=O and C-OH, respectively. After binding, the characteristic peaks of C=O and C-OH shifted to 531.8 eV and 532.6 eV, which proved that carbonyl and hydroxyl oxygen was involved in calcium metabolism [71].

The molecular docking technique (MD) is mainly used to study the interactions between molecules (ligands and receptors). Therefore, it can be used to determine the binding of peptides and calcium ions. Based on MD results, Xu et al. found that calcium ions chelate with Asp residues of a peptide from mussels (LGKDQVRT), demonstrating that an interaction between the carbonyl group and the calcium ion is taking place [72]. In a similar study, Ca^2+^ was found to bind to the Glu carboxylate oxygen of the peptide QEELISK via charge interactions [60].

### 4.3. Nitrogen Atoms

The nitrogen atoms of certain peptides have also been confirmed to bind to metal ions. A peptide (EDLAALEK) derived from sea cucumber eggs was shown to have a superior CaBC. FTIR spectra showed a shift from 3432 cm^−1^ to 3419 cm^−1^ for the corresponding N-H vibrations, indicating that the amino nitrogen atom may have acted as a calcium binding site [73]. Wu et al. found that new bands with wave numbers of 555.07 cm^−1^ appeared following the binding of the peptide to calcium as a result of the stretching vibrations of N-Ca [13]. Furthermore, the binding site of the egg phosphopeptides and calcium ions was determined via NMR hydrogen spectroscopy. The peak corresponding to the amino hydrogen atom at 1.45 ppm in the spectrum was found to have shifted after binding, which indicated that the amino group was a binding site in the chelation reaction [57].

Histidine also has the ability to bind calcium ions due to its nitrogen atom on the imidazole group. For example, Sun et al. found that the peak in the NMR hydrogen spectrum of an egg white peptide (DHTKE) at 14.19 ppm (corresponding to the hydrogen atom of the imidazole nitrogen group) disappeared after binding to calcium ions, which suggested that the nitrogen atom in the imidazole group was one of the binding sites [71]. Similarly, the characteristic signal in the FTIR spectrum of the imidazole group portion of histidine shifted from 1029.16 cm^−1^ to 1030.22 cm^−1^ during binding [13]. Thus, the amount of histidine in a peptide is likely to affect its own CaBC.

### 4.4. Binding of Peptides to Calcium Ions

As shown in Figure 1, the main ways in which calcium ions chelate to the carboxylic acid group (-COO-) on peptides are the unidentate, bidentate, bridging and alpha modes [74]. In the unidentate mode, a metal ion interacts with only one oxygen atom of the -COO- group; when a metal ion interacts with two oxygen atoms on a -COO- group, the mode is termed bidentate; and, if the two oxygen atoms in the -COO- group are each bound to a metal ion, it is the bridging mode. When the calcium ion is chelated with one oxygen atom of the carboxylate group and another ligand atom (O, N, S, etc.), it is the alpha mode A decapeptide from Pacific cod bone was previously reported to have three binding modes with calcium ions, namely the unidentate, bidentate and alpha modes [37].

The binding of the phosphate group to the calcium ion can also occur via three modes, namely unidentate, bidentate and tridentate, depending on the number of phosphate groups or other groups involved in the binding. Luo et al. used molecular dynamics to visualize the binding mode of casein phosphopeptides. The calcium binding sites included the carbonyl group of the Glu residue and the phosphate group of the phosphoserine residue, and all three of the chelation modes were observed. According to the model representation, the phosphopeptides used in the study were bound to calcium ions at a ratio of 1:6 [75].

## 5. The Security and Stability of Calcium–Peptide Chelates

### 5.1. Security

Currently, CaBP can be derived from a wide variety of food sources; however, health and safety have become significant concerns. While CaBPs are extracted from food proteins, it is possible that certain allergic substances such as biogenic amines, D-amino acids and lysine could be formed during protein or peptide processing. By comparing peptide sequences, it has been found that some biologically active peptides have sequences that are identical to those of toxic peptides. For example, the antioxidant peptide Phe-Lys-Lys, extracted from shrimp muscle, is often found in toxic peptides [76]. This phenomenon suggests that biologically active peptides could pose a serious threat to human health, due to their special sequences [77]. Therefore, it is essential that the safety of a particular CaBP be confirmed before it is used as a calcium supplement.

Cells are currently commonly used for the safety evaluation of calcium–peptide chelates. Thiazolyl blue tetrazolium bromide colorimetry (MTT) is commonly used to determine cell viability, in which the main principle is that living cells are able to cause the discoloration of the MTT reagent while dead cells cannot. Huang et al. treated Caco-2 cells with different concentrations of calcium chelates of egg white peptides (EWPs-Ca). Cell viability reached its highest value at 50 μg/mL and increased by 35% compared with the control group, thereby indicating that the EWP-Ca was not cytotoxic [78]. In addition, Lin et al. proved that the viability of Caco-2 cells treated with snapper fish scale protein peptide and CaCl_2_ was greater than 85% [42], and a chelate formed by basil seed peptides and calcium ions showed similar properties [18]. In combination, none of the different sources of peptide chelates with calcium ions have been reported to exhibit cytotoxicity. In current literature, in which the cytotoxicity of peptides and chelates were determined, most have been deemed safe. However, their animal toxicity and allergy sensitization should be fur-ther ascertained in future research to ensure that they will not cause harm to the human body.

### 5.2. Stability

The structure of calcium–peptide chelates is destroyed after ingestion due to digestive effects and their absorption can be affected by the structural changes of the peptide. In addition, calcium ions are competitively bound by dietary oxalic acid, tannins, polyphenols and long-chain fatty acids to form precipitates, leading to decreased solubility and reduced absorption by the body [46]. The stability of calcium supplement products is, thus, a key factor in its full utilization. Currently, methods for evaluating the stability of calcium–peptide chelates include mainly thermal stability analysis and stability analysis under a gastrointestinal environment.

Many compounds undergo mass changes and heat absorption or release during temperature changes. The stability of substances can be determined from their changes in mass and heat. When thermogravimetric analysis was performed on calcium bovine bone collagen peptide chelates (CPs-Ca), the thermogravimetric curve showed that the weight of the CPs decreased rapidly with increasing temperature, while the weight of the CPs-Ca showed only a slow rate of decrease with the increasing temperature. The main reason for this phenomenon may be the formation of new chemical bonds between the calcium–peptide chelates and calcium ions, indicating that CPs-Ca was a more stable substance [40]. A CaBP was extracted from *Schizochytrium* sp. protein, and a differential scanning calorimetry curve showed that its four endothermic peaks were 155.16 °C, 161.26 °C, 298.66 °C and 386.34 °C. After binding with calcium ions, the corresponding temperatures of the four peaks shifted to a higher temperature range, suggesting that the thermal stability improved after the chelation [16]. Lin et al. found that two heat absorption peaks appeared on the DSC curve of the peptide extracted from snapper fish scales. The thermogravimetric (TG) curve showed that the peptide lost 69.94% of its weight during heating, while there was no heat absorption peak on the DSC curve after calcium ion chelation, and the TG curve also showed that the chelate lost 54.31% of its weight, which indicated that the chelate was more stable [42].

To overcome the low bioavailability of calcium, it is necessary to study the stability of calcium–peptide chelates in the human gastrointestinal environment. Wang et al. suggested that the calcium retention of calcium–peptide chelates decreased slightly with an increasing temperature but remained above 80%, indicating that the calcium–peptide chelates were relatively stable within a certain temperature scope. Subsequent digestion with different individual types of proteases showed that the calcium content of the calcium–peptide chelates decreased by approximately 40% in the largest group and by only 13% in the smallest group. The results of this study demonstrated that the calcium–peptide chelates were stable in the gastrointestinal environment [26]. In addition, when the calcium release rate of a chelate formed by whey protein peptides and calcium ions (WPH-Ca) was measured at different pH levels, with CaCl_2_ used as a control, the results indicated that the WPH-Ca chelate maintained a high calcium release rate and prevented the precipitation of calcium ions after being ingested by the body [79].

According to current research, the embedding method can effectively improve the stability of calcium–peptide chelates. Liposomes, which are spherical microscopic vesicles composed of lipid bilayer membranes, are able to encapsulate and slowly release biologically active ingredients. Zhang et al. found that when chelates (egg white peptide and calcium ion) were encapsulated by nanoliposomes, their stability improved. The nanoliposomes slowed the release of calcium during gastric digestion, facilitating the entry of more calcium into the intestine [80]. Microencapsulated particles are widely used for active ingredient delivery due to their good controlled-release properties. Microcapsules formed by the polymerization of chitosan and tripolyphosphate could resist the digestion of a desalted duck egg white calcium–peptide chelate by the stomach [3].

## 6. Absorption Pathways of Calcium–Peptide Chelates

### 6.1. Pathways of Calcium Absorption

The absorption of calcium by the intestinal cells is an important part of its utilization by the body [12]. Therefore, it is necessary to understand the main pathways and key sites of the intestinal absorption of calcium. As shown in Figure 2, the two main absorption pathways of calcium ions in the intestine are transcellular and paracellular [81].

#### 6.1.1. Transcellular Pathway

The transcellular pathway consists of three main steps. First, calcium ions cross the brush border membrane (BBM); next, calcium ions in the cell move from the brush border membrane to the basal lateral membrane; and, finally, the calcium passes through the basal lateral membrane [82]. Three models of transcytotic calcium uptake have been developed, namely facilitated diffusion, vesicular transport and endoplasmic reticulum transport [83].

##### Facilitated Diffusion

Calcium ions enter the enterocyte via the epithelial calcium channels TRPV6, TRPV5 and Cav1.3. After entering, calcium is bound to calcium-binding proteins (synthesis induced by the steroid hormone 1.25 (OH)_2_D_3_) and transported through the cytoplasm to the BLM. Eventually, the calcium is transferred from the calcium-binding protein to the higher-affinity Ca^2+^-ATPase (PMCA1b) or Na^+^/Ca^2+^ exchanger (NCX1), both of which are involved in the excretion of calcium ions from the enterocyte into the bloodstream [84,85]. This is known as an unsaturated diffusion model.

##### Vesicular Transport

A rapid increase in the calcium concentration around the BBM disrupts activity in the filaments in close proximity to the calcium channels and initiates the formation of endocytic vesicles. The newly formed calcium-rich vesicles are transported via microtubules, and some fuse with lysosomes. Finally, the vesicles or lysosomes are transported to the BLM for fusion with the membrane, and calcium ions are extruded into the extracellular medium [83].

##### Endoplasmic Reticulum Transport

This mode of transport utilizes the endoplasmic reticulum, which was first identified in pancreatic cells and has been shown to act as a pathway for the transport of calcium ions by intestinal cells. Calcium ions enter the enterocyte via a store-operated calcium (SOC) channel and are then drawn into the endoplasmic reticulum by the Ca^2+^ pump on the endoplasmic reticulum (SERCA). Release channels IP3R and RyR are concentrated in the part of the endoplasmic reticulum near the basolateral membrane. Calcium ions released from the endoplasmic reticulum are eventually released into the blood via the PMCA [86].

#### 6.1.2. Paracellular Pathway

The paracellular pathway comprises mainly tight junctional structures (TJ) between cells that are specialized membrane domains located between the apical and basolateral membranes of an enterocyte [85]. These junctions consist of transmembrane proteins, cytoskeletal components and cytoplasmic plaques. Transmembrane proteins of the TJ are synthesized in neighboring cells and include zonula occludens-1, occludin and claudin [87,88], all of which are linked between cells and restrict the free movement of material through the paracellular space. According to previous studies, the movement of ions through tight junctions depends to a large extent on the concentration gradient of osmotic ions and the electrical gradient across the epithelium.

### 6.2. Pathways of Peptide Absorption

Calcium–peptide chelates are reportedly able to resist gastric digestion and can enter the intestine as short peptides [26]. Therefore, an understanding of the pathway of the intestinal cell transport peptide is helpful in determining the specific mechanism through which a calcium–peptide chelate promotes calcium absorption. As shown in Figure 3, the main pathways currently known to transport peptides through the intestinal cells are the PepT1 pathway, the cell-penetrating peptide pathway, both of which are transcellular pathways (similar to the calcium ion transcellular pathway), and the paracellular pathway [89].

#### 6.2.1. PepT1 Pathway

The PepT1 pathway is a broadly specific peptide transporter protein able to transport almost all dipeptides and tripeptides. The PepT1 transport pathway is an H^+^-dependent pathway. After the uptake of peptides and H^+^ from outside the cell by the PepT1 protein, intracellular H^+^ leaves the cell via a Na^+^-H^+^ exchange pump on the cell membrane to exchange Na^+^ entry. The Na^+^ is subsequently excreted through a Na^+^-K^+^ pump on the basolateral membrane. Three Na^+^ are transported out of the cell in exchange for two K^+^, thus restoring the electrochemical gradient [90]. Intracellular peptides can be excreted into the bloodstream via other transporter proteins or can remain in the cell for cellular use.

#### 6.2.2. Cell-Penetrating Peptide Pathway

Cell-penetrating peptides can translocate across the plasma membrane while carrying certain substances into the cell [91]. Penetrating peptides have a cationic portion on their surface that can bind to the negative charge on a plasma membrane, thereby causing a change in the structural shape of the plasma membrane [92]. These pathways take two main forms, either direct transport across the cell or transport through vesicles. Cell-penetrating peptides can be used for novel nutrient delivery to overcome the impenetrability of the cell barrier.

#### 6.2.3. Paracellular Pathway

Paracellular peptide pathways also occur through the tightly linked structures between cells. This pathway transports mainly short peptides (mainly 3–5 amino acids) due to the small pore size of the tight junction structures [89]. However, some larger peptides that do not pass through PepT1 can be transported using this pathway after hydrolysis.

## 7. Bioavailability of Calcium–Peptide Chelates

### 7.1. In Vitro Bioavailability

HT-29 and Caco-2 cells feature most commonly as research objects in recent experiments on calcium absorption. One theory as to how calcium casein phosphopeptide chelate (CPP-Ca) promotes calcium uptake is that CPP may insert itself into the plasma membrane and form its own calcium-selective channel [65]. The reason for this phenomenon may be that CPP has a similar function to that of cell-penetrating peptides. Further studies have shown that this promoting effect of CPP-Ca relies mainly on a phosphorylated acidic motif and a special structure at the N-terminus, characterized by the presence of a loop structure and a β-turn, which may facilitate its insertion into the cytoplasmic membrane. It is noteworthy that the single acidic motif has not shown calcium absorption capacity [93]. In a related study, CPP-Ca was dramatically more bioavailable than calcium chloride and calcium L-aspartate by significantly increasing the expression of TRPV6 and TRPV5, further supporting the idea that calcium–peptide chelates can improve cellular calcium uptake [35].

In addition to casein phosphopeptides, chelates formed by peptides from other sources with calcium ions also promote calcium uptake by cells. For example, it was shown that cells treated with calcium fish scale protein hydrolysate chelates (FSPH-Ca) increased calcium transport by 60.43%, and the mRNA level of the TRPV6 protein was significantly increased. However, further research showed that 2-aminoethyl diphenylborate (2-APB, an inhibitor of TRPV6) was able to slow this facilitation. These results suggest that FSPH-Ca can promote calcium entry into Caco-2 cells by regulating the opening of TRPV6 channels and mRNA expression [42]. Wu et al. utilized a Caco-2 cell monolayer model to determine the amount of calcium transported. It was indicated that a porcine osteopeptide calcium chelate could significantly enhance calcium transport [34]. Another recent study found that soybean peptides with high CaBC could transport calcium into cells for absorption [94]. Moreover, the conformation of the peptide was changed during the interaction of the membrane, which was beneficial to the transmembrane of the calcium–peptide chelates.

The interaction of CaBP with calcium transport proteins was simulated by MD. Wu et al. modeled the interaction of six peptides (TSSLEK, YDEVSST, SSVSLELC, KMKSSEN, KEGKEVTFQT, SEKMKILELP) with TRPV6. The respective binding sites show that hydrogen bonding and hydrophobic interactions play an important role in the binding of peptide to TRPV6. In addition, the acidic amino acid Glu has a high CaBC, and the exposed Glu carboxyl oxygen can chelate calcium ions after docking. In contrast, the docking results of SSVSLELC with TRPV6 indicate that a continuous serine sequence can facilitate calcium transport when the serine is located at the nitrogen terminus [95]. This facilitates the entry of calcium ions into the intestinal epithelium.

### 7.2. In Vivo Bioavailability

The everted rat intestinal sac absorption model, used in studies of nutrient absorption effects, was used to measure the absorption of calcium krill peptide chelates (QEELISK-Ca) in a study by Sun et al. The results showed that the absorption of QEELISK-Ca in the duodenum and the ileum was significantly higher than that of CaCl_2_, thus demonstrating that QEELISK-Ca had better absorption compared with inorganic calcium salts [60]. Similarly, the absorption of calcium egg white peptides chelates from was 32.38 µg/mL, which was significantly higher than that of CaCl_2_ [96].

Animal models have been widely used to investigate the bioavailability of calcium supplement products in vivo. In the study by Lv et al., four-week-old male and female SD rats were divided into three groups and fed with either calcium soy-peptide chelates (SPH-Ca group), a casein phosphopeptides calcium complex (CPP-Ca group) or calcium lactate (L-Ca group). The results indicated that the femoral bone mineral density and lumbar spine loading capacity in rats of the SPH-Ca and CPP-Ca groups were significantly increased [31]. Serum calcium levels (SCL) can serve as a valid indicator of whether an active substance has a pro-calcium absorption effect. For example, SCL in mice were found to be significantly increased after they were administered calcium phosphopeptide chelates [97]. Similar results were found in a study about the bioactivities of calcium herring egg phosphopeptides chelates.

Low-calcium diet rats are widely used in the study of calcium utilization. When rats on a low-calcium diet were fed calcium chlorella peptide chelates (CPPH-Ca), the results showed that CPPH-Ca increased bone mineral density (BMD) and bone mineral content (BMC) while decreasing serum alkaline phosphatase (ALP) and inhibiting morphological changes in bone. The findings led the authors to suggest that CPPH-Ca significantly improved the skeletal condition of mice [15]. In a study of calcium tilapia fish scale peptide chelates (TSPH-Ca), different groups of hypocalcemic rats were fed either normal saline (control), calcium carbonate (Ca-CO_3_), TSPH-Ca or casein phosphopeptide-calcium (CPP-Ca). The results of physiological indexes showed that compared with the parent calcium absorption, calcium contents in the serum and femur and femoral bone density and strength were significantly higher in the rats fed TSPH-Ca than in the control and CaCO_3_ groups [59].

The ovariectomized rat (OVX) is another commonly used research model due to its symptoms of osteoporosis. Zhang et al. investigated the anti-osteoporotic activity of calcium cod bone peptide chelates with the OVX model. CBP-Ca was suggested to significantly increase bone biomechanical properties, bone mineral density and bone mineralization, as well as significantly reducing the serum levels of osteocalcin, bone alkaline phosphatase, TRAP5b and the type I collagen carboxy-terminal cross-linked peptide, resulting in healthier bones in the rats [38]. In another study, OVX rats were fed bovine collagen peptides and calcium citrate. It was found that the bone conditions of the rat in the experimental group was significantly better than those of the normal diet group [98].

Overall, calcium–peptide chelates have been shown to effectively increase the bioavailability of calcium in cells and improve bone condition in rats. These findings suggest the potential of calcium–peptide chelates as calcium supplements, which could effectively meet people’s demand for calcium.

## 8. Conclusions

This paper summarizes the research on calcium–peptide chelates. It was shown that the hydrolysis and purification techniques used for the preparation of CaBP from different food-derived proteins are relatively comprehensive. The peptides have many active sites and can bind calcium ions in different modes. Furthermore, biological experiments have shown that calcium–peptide chelates can significantly improve calcium utilization in cells and rats.

The study of calcium–peptide chelates is meaningful in light of the importance of calcium to the human body. Future research should prioritize first the development of new techniques for the preparation of CaBP; second, the determination of CaBP sequence characteristics; and, third, the clarification of the absorption pathways and mechanism of action of calcium–peptide chelates.

## Figures and Tables

**Figure 1 foods-11-02762-f001:**
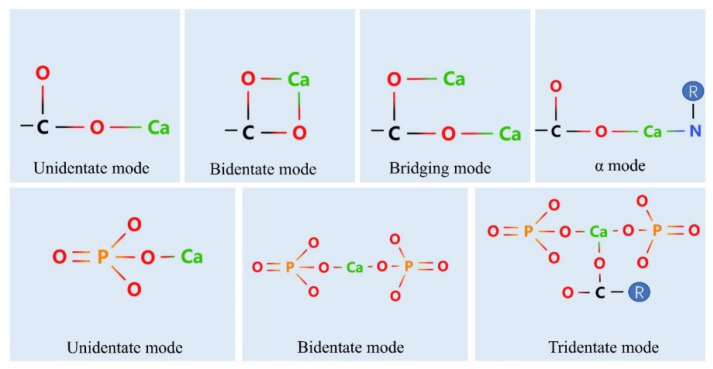
Binding modes of calcium ions to active sites.

**Figure 2 foods-11-02762-f002:**
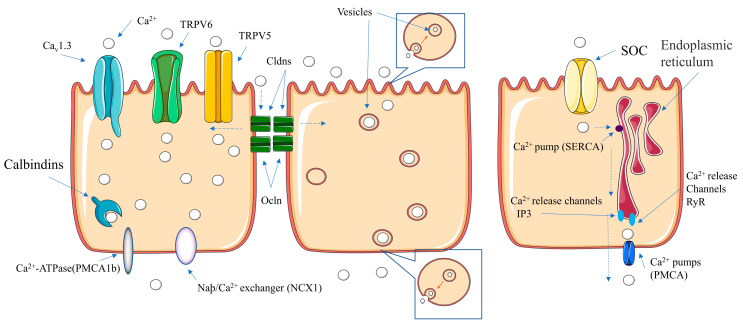
Absorption pathways of calcium ions in the intestine.

**Figure 3 foods-11-02762-f003:**
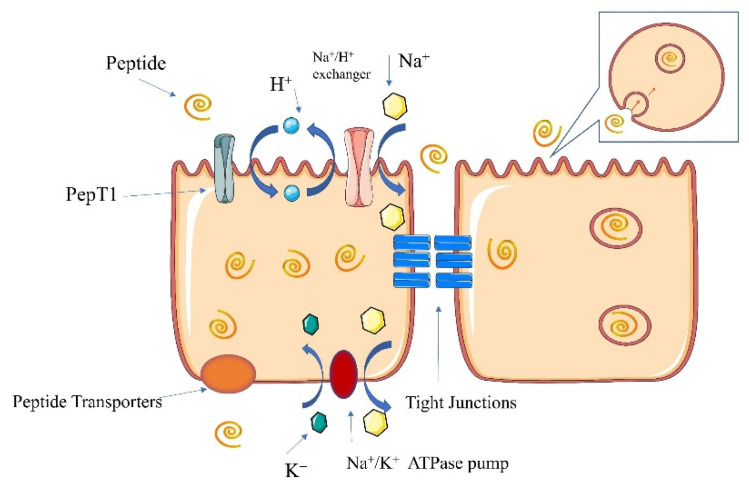
Absorption pathways of peptides in the intestine.

**Table 1 foods-11-02762-t001:** Peptides with calcium binding capacity (CaBC) and their various sources.

Source	Preparation	Amino Acid Sequence	CaBC (Ca^2+^/Peptide)	Ca^2+^ Binding Site
Chlorella (*Chlorella vulgaris*) [15]	Flavourzyme hydrolysis	NSGC	211 µg/mg	Ser, Gly and Cys residues
*Schizochytrium* sp. [16]	Alcalase and Flavourzyme hydrolysis	FY	128.77 µg/mg	Carboxyl oxygen atoms and amino nitrogen atoms; nitrogen and oxygen atoms of amido bonds
Cucumber seed [17]	Liquid state fermentation with *B. subtilis*		191.5 µg/mg	-COOH, -OH, -NH_2,_ -CO-NH-
Lemon basil seeds (*Ocimum citriodorum)* [18]	Alcalase hydrolysate	AFNRAKSKALNEN, YDSSGGPTPWLSPY	278.14 µg/mg, 151.88 µg/mg	Amino nitrogen atoms and oxygen atoms on the carboxyl group
Soybean [19]	Protease M&Amano enzyme	DEGEQPRPFPFP	79.24 μg/mg	Glu, Gln, Lys and Pro
Black bean [20]	ficin hydrolysate		77.54 μg/mg	Amino nitrogen atoms and carboxyl oxygen atoms
Mung bean [21]	Enzymatic hydrolysis	LLLG, AIVIL, HADAD	943.60 μg/mg, 834.87 μg/mg, 809.13 μg/mg	Leucine or isoleucine at the C- or N-terminal
Alaska pollack (*Theragra chalcogramma*) backbone [22]	Pepsinolytic hydrolysis	VLSGGTTMAMYTLV	160.00 μg/mg	
Tilapia [23]	Alcalase (2.4 L) hydrolysis	WEWLHYW	65 μg/mg	
Bovine serum protein [24]	Alcalase, Flavourzyme and Protamex hydrolysis	DNLPNPEDRKNYE	14.18 μg/mg	
Frigate mackerel (*Auxis thazard*) [25]	Enzymatic hydrolysis and membrane separation	EPAH	76.8 ± 4.5 μg/mg	Carboxylic group of Glu, carboxylic group and the amino group of His
Sheep bone [26]	Enzymatic hydrolysis and *Lactobacillus* fer-mentation		56.39 μg/mg	Carboxyl oxygen and amino nitrogen atoms of collagen peptides
Porcine blood [27]	Flavourzyme hydrolysis	VSGVEDVN	7.758 μg/mg	
Shrimp processing by-products [28]		TCH	11.76 μg/mg	
Akiami paste shrimp (*Acetes japonicus*) [29]	Flavourzyme	YEIPAEDL	277.96 ± 20.93 μg/mg	Carboxylate oxygen of Glu and Asp

## Data Availability

Not applicable.

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
