# Peer review of "The Formation, Structural Characteristics, Absorption Pathways and Bioavailability of Calcium–Peptide Chelates"

_foods, 2022, doi:10.3390/foods11182762_

Round 1

Reviewer 1 Report

The authors present an interesting review that provides a relevant insight into the calcium-peptide chelates. The introduction of the manuscript is well prepared and follows the theme of the article. The figures and tables are self-explanatory, clear, and easy to understand. However, some comments should be addresed. 

1. L48-L49. Please add more important characteristics about peptides. 

In the follow article can you find more characteristics at the third paragraph. Peptides 122 (2018) 170170.

2. Are there some in silico or bioinformatic studies about this kind of peptides? If yes, please add information

Reviewer 2 Report

  • It would be interesting if the units in which the CaBC is expressed in Table 1 and throughout section 2 were unified, in order to be able to compare them

  • In the line 552 appears:”……diferent food-proteínico are relatively mature”. I think that a mature synonym could be put that would be more appropriate for the meaning of that phrase

  • revise the text so that "calcium peptide chelates" is always written the same way ( ejem, lunes 550, 554 y 556)

Reviewer 3 Report

it is a good peper, well written that gives inspiration for other works. it is written in correct English language
